# A Rapid Realist Review of Group Psychological First Aid for Humanitarian Workers and Volunteers

**DOI:** 10.3390/ijerph18041452

**Published:** 2021-02-04

**Authors:** Julia Corey, Frédérique Vallières, Timothy Frawley, Aoife De Brún, Sarah Davidson, Brynne Gilmore

**Affiliations:** 1UCD Centre for Interdisciplinary Research, Education and Innovation in Health Systems, School of Nursing, Midwifery & Health Systems, University College Dublin, Dublin 4, Ireland; julia.corey@ucd.ie (J.C.); timothy.frawley@ucd.ie (T.F.); aoife.debrun@ucd.ie (A.D.B.); 2Trinity Centre for Global Health, Trinity College Dublin, Dublin 2, Ireland; fvallier@tcd.ie; 3British Red Cross, London EC2Y 9AL, UK; SDavidson@redcross.org.uk

**Keywords:** group psychological first aid, psychological first aid, humanitarian workers, psychosocial support

## Abstract

Humanitarian workers are at an elevated risk of occupational trauma exposure and its associated psychological consequences, and experience increased levels of anxiety, depression, and post-traumatic stress disorder (PTSD) compared to the general population. Psychological first aid (PFA) aims to prevent acute distress reactions from developing into long-term distress by instilling feelings of safety, calmness, self- and community efficacy, connectedness and hope. Group PFA (GPFA) delivers PFA in a group or team setting. This research sought to understand *‘What works, for whom, in what context, and why for group psychological first aid for humanitarian workers, including volunteers?’* A rapid realist review (RRR) was conducted. Initial theories were generated to answer the question and were subsequently refined based on 15 documents identified through a systematic search of databases and grey literature, in addition to the inputs from a core reference panel and two external experts in GPFA. The findings generated seven programme theories that addressed the research question and offered consideration for the implementation of GPFA for the humanitarian workforce across contexts and age groups. GPFA enables individuals to understand their natural reactions, develop adaptive coping strategies, and build social connections that promote a sense of belonging and security. The integrated design of GPFA ensures that individuals are linked to additional supports and have their basic needs addressed. While the evidence is sparce on GPFA, its ability to provide support to humanitarian workers is promising.

## 1. Introduction

### 1.1. The Problem

Humanitarian workers often work in difficult contexts such as disease outbreaks, natural disasters, and political conflicts [1], putting them at an elevated risk of occupational trauma exposure and its associated psychological consequences [2]. Increased risks of adverse mental health outcomes among humanitarian aid workers have been well documented by a range of studies spanning the last two decades. For example, in reviewing the evidence for trauma exposure and trauma-related mental illness among humanitarian aid workers, Connorton et al. [2] found that humanitarian workers experience increased levels of anxiety, depression, and post-traumatic stress disorder (PTSD) compared to the general population. Similarly, an increased risk of anxiety and depression symptoms have also been reported among aid workers after controlling for the presence of such disorders prior to deployment, providing evidence for the experiential nature of humanitarian missions as a mediating factor in mental health outcomes [3].

Further differentiating between expatriate (i.e., those hired outside of the country of mission) and national staff (i.e., those hired within the country of mission), Lopes Cardozo et al. [4] found that national staff reported more exposure to traumatic events as well as greater anxiety, depression, and PTSD symptoms, compared to expatriate staff. Similar findings were reported by Musa and Hamid in Darfur [5]. Having shared experiences and cultures, national staff often identify more closely with the victims than their expatriate counterparts [4,5,6]. Despite an arguably greater need for psychological support among national staff due to increased exposure and adverse symptoms, however, extant studies on traumatic exposure and mental health among humanitarian aid workers have overwhelmingly focused on expatriate or international staff [4,7]. Consequently, there has been considerably less focus given to national staff and volunteers, the latter of which represent a significant proportion of the humanitarian workforce. One report for example, suggests a ratio of 1:180 staff to volunteers within humanitarian organisations in low and middle-income countries [8]. Volunteers and national staff thus appear more likely to face elevated risks of depression and anxiety, while simultaneously receiving less organisational support [4,6,7].

Moreover, younger humanitarian aid workers and volunteers (both expatriate and national staff) are at a greater risk of psycho morbidity [6] and experiencing burnout than their older counterparts [5,9], even when controlling for years working within a humanitarian agency [9]. Beyond the significant adverse impact at the individual level, the consequences of a negatively affected health workforce are not only detrimental to humanitarian efforts and humanitarian organisations themselves, but also pose a risk to the efficacy and quality of humanitarian interventions [10,11].

Humanitarian organisations that respond to the needs of volunteers, through supportive practices and skill-building approaches, for example, have been found to enhance motivation and well-being among volunteers [12,13]. These psychological states are documented as important determinants of workforce retention and performance [14]. Accordingly, an increasing number of humanitarian organisations are investing in resources and mounting efforts to improve support and well-being among their workforce, including volunteers. One such organisation is the International Federation of Red Cross and Red Crescent Societies (IFRC), which recently published their *Caring for Volunteers* psychosocial support tool kit [15], as a way to equip volunteers with the skills they need to ensure that they are caring for their own mental health and well-being, in addition to the health and well-being of the populations they serve. 

### 1.2. Group Psychological First Aid

Psychological first aid (PFA) evolved as an alternative to psychological debriefing [16], an intervention that involves sharing details of traumatic experiences and emotions [17]. Multiple studies have shown that psychological debriefing does not improve recovery from psychological trauma [17], and in some cases, may actually negatively impact mental health outcomes [18,19]. In contrast, PFA does not involve discussions about the recent traumatic event, but instead focuses on providing humane, supportive, and practical help to individuals who are suffering and in need of support [20]. As an overarching supportive approach responds to the urgent physical and psychological needs, PFA can be used immediately in the aftermath of a traumatic experience as well as in the days or weeks afterwards. PFA can also be used within programmes where humanitarian workers are exposed to prolonged and chronic stressors (i.e., during a protracted crisis), and aims to prevent acute distress reactions from developing into longer-term distress [21].

The main purpose of PFA is to instil feelings of safety, calmness, self- and community efficacy, connectedness and hope; elements deemed ‘essential’ to trauma interventions in the early aftermath of disasters and mass violence by Hobfoll and colleagues [22]. Provided that basic physical needs are also being met within a humanitarian response, PFA thus works to meet the psychological needs of individuals through providing comfort and support, psychoeducation, and facilitating service connections to continued mental health resources [23]. The three main principles of PFA are to *look* (for safety, for who needs help), *listen* (to the person in distress) and *link* (to further support) [20,24]. Despite a dearth of studies examining the effectiveness of PFA [25], PFA is widely used within the humanitarian sector, including during disease outbreaks and pandemics [16,20,21,24,26,27,28,29]. While ascertaining the effectiveness of PFA remains particularly challenging, PFA is recognised as being evidence-informed [30], and has been shown to improve our knowledge and understanding of psychological response and skills in providing support to those exposed to acute adversity [31].

Group PFA (GPFA), delivered in a group or a team setting, is a more recent adaptation of PFA that is supported by several major agencies, including the IFRC, as an effective way to care for staff and volunteers in crisis [21]. As exposure to trauma can be extremely isolating for individuals [32], the provision of psychoeducation and PFA in a group setting can help normalise reactions and responses to trauma and strengthen group cohesion [23]. According to Eriksson et al. [9], organisational support and positive relationships with co-workers may also increase resilience among staff. Like PFA, the provision of GPFA is not only limited to professional counsellors but can be provided by trained workers, volunteers and peers [20,24]. GPFA therefore offers humanitarian organisations the opportunity to provide an important resource to staff and volunteers, as a likely lower-cost, scalable, and potentially highly effective mental health and psychosocial support initiative, which can be delivered by managers to humanitarian workers before, during and after responding to crises. In addition, GPFA also has the potential to build peer support networks within a team [23,24]. The implementation of GPFA, including when it is initiated, by whom and how often, is often at the organisation’s discretion and dependent on the situation. 

Given the increasing recognition of the importance of supporting staff and volunteers’ mental health within crisis settings [10], it is likely that GPFA will continue to attract increased attention in the coming years. Extant literature on the potential impact of GPFA to prevent or address anxiety and/or depression in the workplace, however, remains scarce. Therefore, the current study aimed to draw from the available evidence, including theoretical frameworks available from the existing PFA literature as well as similar group-style psychosocial based interventions, to understand “What works, for whom, in what context, and why for Group Psychological First Aid for humanitarian workers, including volunteers?”. In addition, and given that the humanitarian workforce are largely comprised of volunteers and staff who are at an early stage of their career, and that 75% of all lifetime mental health problems occur by the age of 24 [33], we further sought to understand how GPFA may be particularly relevant to young workers and volunteers (defined as those aged under 25).

## 2. Materials and Methods

A rapid realist review (RRR) was conducted to answer the research question, “What works, for whom, in what context, and why for Group Psychological First Aid for humanitarian workers and volunteers?” Realist methodology is a useful approach for understanding the evidence for complex interventions, because it examines how and why interventions work under different contexts [34]. Specifically, a realist review approach aims to understand what works, for whom, under what circumstances, and why, through examining generative causation, or how interactions between contextual factors (C) and underlying mechanisms (M) generate outcomes (O) given a specific intervention [35]. To do so, it elicits and refines programme theories that explain how, why and for whom the intervention works (or does not) by uncovering generative causation, often expressed as the heuristic tool of ‘context-mechanism-outcome’ configurations (CMO configurations) (that is, contexts trigger mechanisms, and together this combination generates outcomes). It is these CMO configurations (CMOCs) that provide the support to both generate, and then test/refine programme theories that answer the research question. Guided by Pawson and Tilley [36], this review disaggregates mechanisms into ‘resources’ and ‘responses’, which as noted by Dalkin, supports distinguishing between the key, and the often difficult to conceptualize concepts of contexts and mechanisms [37]. This RRR follows the RAMESES (Realist and Meta-narrative Evidence Synthesis: Evolving Standards) realist publication standards guide for realist syntheses [38], with adaptations made to accommodate the inclusion of a reference panel and external experts, as outlined below. 

Whereas traditional realist reviews require both considerable time and investment [34], an RRR maintains the same core principles and approach while streamlining the review process through engaging experts in the field of study, often called a reference panel [39]. The literature included in RRRs is therefore not intended to be exhaustive, but rather to represent the most relevant and informative resources. In this way, RRRs are particularly useful and relevant to policy-makers and stakeholders facing time-sensitive decisions [39]. This methodology also allows the application of existing theories and evidence to make inferences as to how programmes are expected to work, which can be useful for topics with limited evidence bases [39], such as GPFA. Figure 1 outlines the key research steps throughout this RRR. As highlighted, RRRs start and end with a theory, which are tested and refined throughout the review process through both the CMOCs elicited from the literature and the reference panel’s guidance and inputs. 

### 2.1. Reference Group and Experts 

Another key distinguishing factor of RRRs is the involvement of a reference panel [39]. For this review, we established a Core Reference Group which guided the development of our research question, protocol and who provided feedback on the theory development throughout the study. The panel consisted of an academic with expertise in mental health service delivery, and two programme implementers with extensive experience in designing and implementing workplace mental health interventions within non-governmental organisations (NGOs), including PFA and GPFA. We also consulted two External Experts, one of which works for a large NGO, and the other for a UN organisation, both designing, training and implementing GPFA across a wide variety of humanitarian contexts, including conflict, natural disasters and health/disease emergencies. These experts were consulted towards the end of the review for the validation of findings and to elicit their insights into how these findings might inform recommendations for future practice. 

### 2.2. Preliminary Review of the Literature and Initial Programme Theories

The research team began by reviewing several pieces of key literature [20,21,23,24,28,29,40] on PFA and GPFA to extract CMOCs and to gain a broader understanding of how and when these approaches are used within humanitarian contexts. These CMOCs were then synthesised to develop candidate theories for how GPFA may work to prevent and/or address anxiety or depression in the workplace. After receiving feedback from the reference panel on the candidate theories, the research team returned to the literature for further exploration and revisions to develop initial programme theories (IPTs). These IPTs were reviewed again by the reference panel before proceeding to develop the systematic literature search.

### 2.3. Systematic Literature Searching

The systematic searching strategy for relevant literature was agreed upon by all members of the research team and core expert panel, and consisted of database searching, grey literature searching and snowballing. The searching of three academic databases (PubMed, Scopus and Taylor and Francis Online), seven websites of relevant organisations (World Health Organization (WHO), United Nations, Mental Health and Psychosocial Support (MHPSS) Network, Relief Web, Elrha, International Committee of the Red Cross (ICRC), and IFRC), emailing to MHPSS listservs requesting documents, and snowballing of included studies’ references were all completed.

Appendix B details the specific search terms used, the dates of searching and returns for each source and inclusion and exclusion criteria applied to the returned documents. Due to the overall dearth of literature on GPFA for the humanitarian workforce, articles that broadly applied to GPFA, such as those related to PFA, GPFA outside of humanitarian workers, and those that described other group-based events for humanitarian workers and first responders were included. While inclusion criteria were quite broad in terms of context, study type, or literature type, included resources must have provided sufficient depth and relevance to contribute to our theory refinement. In general, however, with some exceptions for articles that had strong relevance and richness, resources were to meet at least three out of four of the following criteria in order to be included:About (or applicable to) PFA/GPFA;Provided in a humanitarian or emergency response;Targets youth;Targets anxiety and/or depression.

Once results were returned, a minimum of two investigators screened each resource for relevance. Any disagreements were settled by a third review and/or group consultation.

### 2.4. Changes to Process—Youth Inferences

Our search strategy was originally designed to capture literature focusing on the implementation of GPFA among those aged 14–24. However, no identified resources addressed GPFA or PFA specifically for young populations. To overcome this issue, and following the completion of the RRR with the included studies for a more general understanding of “how, why and for whom GPFA works for humanitarian workers and volunteers”, we further reviewed youth/adolescent-specific literature and made inferences about how the findings from the RRR might be relevant to this population (Figure 1). Consequently, findings applicable to the general population of humanitarian workers and volunteers are presented in the results section, and the relevance of these findings to the young workforce is included in the discussion section. 

## 3. Results

### 3.1. Data Analysis and Research Activity

As highlighted in Figure 1, the findings of this review emerged from the analysis of a number of data sources, in addition to the following research outputs and activities:The inclusion of 15 documents identified through a systematic search of databases (*n* = 6), websites, and grey literature (*n* = 9) (See Figure 2);Two reference panel virtual meetings, where theories were presented and feedback provided;Two feedback reports from the reference panel assessing how input was incorporated, and providing updated findings (programme theories) for their review;Review of literature specific to youth/adolescents participating in group interventions and/or youth/adolescents mental health considerations for interventions;Two virtual meetings with external experts, where the refined programme theories (PTs) and findings from the youth-related supplemental work were disseminated. Inputs and recommendations were sought, and changes were made to produce finalised programme theories.

### 3.2. Defining Group Psychological First Aid

Implementation of PFA/GPFA or similar programmes were examined across the included documents, with differences emerging across specific components of the programmes. To ensure that our findings resulted from and supported high-quality, ethical and evidence-based GPFA, the following expert-group-informed definition and overview of best practice for GPFA was used:

GPFA is a focused, non-specialised support [41] provided to a group of individuals that have collectively experienced an acute stressor (e.g., natural disaster, violent attack, accident) or who are currently experiencing a period of protracted stress (e.g., an ongoing conflict, persistent threat of violence) [23], ideally as soon as possible and where appropriate [21]. GPFA should serve as an entry point for access to a wider system of supports and other resources, whenever they are available [21,23]. Like PFA, the three main principles of GPFA are to look (for safety, for who needs help), listen (to the person in distress), and link (to further support) [20,24]. In doing so, GPFA aims to instil feelings of safety, calmness, self- and community efficacy, connectedness and hope [22], by providing individuals with coping strategies and skills, and facilitating relationship building among group members [23]. GPFA should ideally be led by two facilitators and conducted more regularly (as appropriate and feasible in a given context) in cases of prolonged or more chronic stress [21]. Facilitators do not need to be mental health professionals, however, they must be properly trained in group facilitation skills, in order to effectively provide PFA in a group format [21]. Careful consideration must also be given to group composition (i.e., in terms of gender, age, education level, and other forms of social hierarchies), in line with prevailing social norms with different cultural contexts [24]. 

### 3.3. Theory Refinement 

Programme theories related to the research question underwent four phases of refinement as a result of the literature searches and data extraction, as well as feedback from both the reference panel and external experts. The progression of the finalised programme theories throughout the review process is illustrated in Figure 3 below.

### 3.4. Finalised Theories

As illustrated in Figure 3, six initial programme theories (Appendix C) evolved into a total of seven finalised Programme Theories to address the research question. Revision took place at each step. The included academic and grey literature in support of the development of each programme theory is cited within, with Appendix A detailing the CMOCs and their data sources that contributed to the development of each theory.

#### 3.4.1. Programme Theory 1: Natural Reactions and Adaptive Coping Strategies

Programme theory 1 is supported by 11 CMOCs extracted from 10 literature sources. The theory focuses on participants’ natural reactions to abnormal events, and the importance of learning adaptive coping strategies to cope with the trauma exposure. 


*Following an acute crisis or period of prolonged distress, if GPFA is delivered early and appropriately, it provides a space to discuss natural reactions, normalise relationships, and address expectations. If this occurs, participants will be better equipped to process experiences early, feel their reactions are normal, and to continue supporting their mental health. This can lead to improved healthy coping strategies, self-awareness, and the management and prevention of distress escalation or re-escalation [42,43,44,45,46,47,48,49,50,51,52].*


The nature of humanitarian work often results in the workers and volunteers prioritising others’ well-being before their own [48]. Members of our expert panel agreed that the group format of GPFA may encourage this workforce to attend to support their peers, and through their attendance, they may also receive benefits. GPFA provision can help to facilitate the recognition of stress reactions as natural and reduce hesitancy toward care-seeking among the workforce [42,46,47,48,50]. The provision of these practical tools and dedicated time and space may be especially important in settings with limited or disrupted services.

#### 3.4.2. Programme Theory 2: Meeting Basic Needs

Six CMOCs extracted from five literature sources support programme theory 2. This theory relates to meeting the basic physical (i.e., food, water, shelter, etc.) and psychological (i.e., comfort, sense of stability, etc.) needs of participants.

*Acute crises and periods of prolonged distress affect individuals differently depending on exposure levels or previous life experiences. Basic resource needs may be disrupted, requiring different levels of physical and psychological support. If GPFA is provided in a comfortable location, complemented by a layered system of supports, this can help meet an individual’s basic physical and/or psychological needs, which can increase their sense of stability, safety, and control. If this occurs, individuals are able to be more emotionally expressive, self-efficacious, and recognise their reactions as natural. This helps to prevent distress escalation through emotional stabilisation, reduced secondary stressors, and helping individuals cope on their own [42,43,48,51,52]*.

In an ideal situation, basic needs would be met prior to, or identified during GPFA and addressed through other resources. However, practical challenges may arise, particularly in meeting physical needs, based on contextual factors [44,52]. This is especially true in low-resource settings [42,44]. Our external experts emphasised the importance of setting clear expectations prior to the GPFA meeting to protect the well-being of both beneficiaries and facilitators. 

#### 3.4.3. Programme Theory 3: Response Matched to Individual Needs

Programme theory 3 is supported by 12 CMOCs extracted from 10 literature sources and highlights the importance of recognising individual needs within the group, and providing those members with needs-matched services.


*Acute crises and periods of prolonged distress affect individuals differently depending on exposure levels or previous life experiences. When there is an existing social support/resource system, if GPFA is linked with this system and facilitators can gauge individual reactions and needs, this can enable an open space for members to share reactions, coping strategies, and resources. This can also support individuals to be referred for additional support for them (either through more formal service connections or informal one-on-one meetings with a facilitator). Supporting service connections in group formats can reduce stigma associated with care-seeking and increase access to needs-matched services, reducing secondary stressors and strengthening and individual’s ability to cope on their own [42,43,44,46,47,49,50,51,52,53].*


#### 3.4.4. Programme Theory 4: Fostering Support and Social Cohesion

Programme theory 4 is supported by 10 CMOCs extracted from eight literature sources. This theory focuses on fostering support and social cohesion in two different situations—when group members already know each other (4.1) and when they do not already know each other (4.2).


*4.1—When GPFA is provided to a small group of individuals who already know each other and share similar experiences and stress levels, it provides an opportunity to discuss reactions and emotions. This helps individuals feel more supported by peers, strengthens group cohesion, validates reactions, and opens communication about mental health. The outcome is the fostering of relationships, reduced isolation, increased sense of safety and belonging, and improved coping [42,43,44,46,47,48,50,51,52,54].*


GPFA can be provided to members who do not know each other beforehand. However, an icebreaker and/or group activity aligned with the culture should take place prior to commencing GPFA [43,46]. This approach may be particularly useful in circumstances where individuals are disconnected or separated from their families or communities [52]. 


*4.2—When GPFA is provided to a small group, who may not know each other but share similar experiences and stress levels, and members can forge new relationships with one another, it can develop and strengthen a sense of belonging to communities, foster relationships, communication, and help group members feel less isolated [42,43,44,46,47,48,50,51,52,54].*


The group nature of GPFA may be particularly important in collectivist cultures that naturally feel comfortable in group settings [42,43,46,54]. Offering additional informal or social group spaces beyond GPFA may support ongoing dialogue and support among members. Online platforms such as WhatsApp or Facebook groups can also be useful in events where social gatherings are prohibited or interrupted, such as epidemics or disease outbreaks, or where social gatherings present a security risk [44]. 

#### 3.4.5. Programme Theory 5: Group Composition

Nine CMOCs extracted from nine included documents support programme theory 5. This theory addresses the composition of the group, including both members and facilitators.


*When people experience a similar acute crisis or prolonged stress, and GPFA is provided with members and facilitators working at similar levels (or if different, where neither group holds direct authority over the other), power imbalances can be reduced, supporting open and honest sharing. This can also develop a sense of comradery and group cohesion, increasing communication, participation, and attendance [26,42,44,46,47,48,50,51,54].*


It was unanimously agreed among all our experts that approaches to balancing power dynamics are heavily context specific. Such considerations are especially important in cultures with strong social, gender, and/or age hierarchies, where heterogeneity may influence individuals’ participation in the group session. This may also be relevant in humanitarian workforces where national and local staff come from diverse backgrounds and cultures. 

#### 3.4.6. Programme Theory 6: Sustainability and Accessibility

Programme theory 6 is supported by eight CMOCs extracted from six literature sources and addresses how GPFA can be accessible and sustainable for members:


*Following an acute crisis or period of prolonged distress, when GPFA is linked into complementary supports safely and with cultural competence, GPFA can improve support and access to these support services through increasing visibility and reducing stigma associated with care-seeking. The outcome may be more individuals accessing ongoing support, reduced secondary stressors, decreased isolation, and more sustained support [26,42,43,44,52,54].*


These linkages may be especially important in contexts where individuals may be displaced or have lost access to prior services [26,42,43,44,52,54]. In some cases, the skills gained from attending previous GPFA sessions have supported the early management of reactions among previous participants experiencing new or re-occurring trauma responses. Schafer and Snider (2016) provide an example from a mother, who recounted using the skills she learned in PFA to help her son to calm down, breathe, and evacuate their home just hours before it was destroyed [26]. Individuals who have previously received GPFA may be more likely to seek out support or additional GPFA for themselves or others. 

#### 3.4.7. Programme Theory 7: The Facilitators

Programme theory 7, supported by nine CMOCs extracted from eight included documents, focuses on the facilitators and their training.


*Acute crises and periods of prolonged distress affect individuals differently depending on exposure levels or previous life experiences. When GPFA is provided by appropriately trained facilitators (ideally two), who are able to gauge individual reactions and needs, groups can be composed based on similar distress levels or needs. A second facilitator supports severely distressed members by taking them aside and providing or linking them to more specialised care. This protects individuals’ dignity, reduces the exposure of the member to secondary trauma, and ensures needs-matched care [43,44,46,49,51,52,53,54].*


Facilitators must be trained not only to provide GPFA, but also in how to work with the demographics of each group. Training should be as comprehensive as possible and involve practice or role playing. Facilitators should be accessible for the continuation of care through further informal, interpersonal discussions, particularly in low-resource settings, and/or through linking to ongoing support, where available. Depending on the context, it is also important that the facilitator is relatable to the members. Facilitators themselves also need supportive supervision linkages to additional services, and/or GPFA, to ensure their own self-care and support their own well-being. 

## 4. Discussion

As previously noted, GPFA and PFA evolved as an alternative to psychological debriefing [16]. Whereas psychological debriefing involves sharing details of traumatic experiences, PFA focuses on assessing needs and providing the support and tools necessary to help individuals to cope on their own [20]. Because GPFA provision is informed by the culture and context within which it takes place, it is considered a practical and highly adaptable approach to providing psychosocial care and support to staff members working in humanitarian contexts [55]. While it was beyond the scope of this review to identify evidence demonstrating the effectiveness (or ineffectiveness) of PFA or how its effectiveness may be optimised, across the literature reviewed and among the experts consulted, the consensus was overwhelmingly strong in favour of PFA and GPFA [56].

Moreover, the findings of this RRR offer important insights into the contextual conditions that can affect how, why, and for whom GPFA works for humanitarian workers/volunteers. These include the implementation conditions in which GPFA is established, the supports and relevant linkages available to participants, the connectedness of the group and their ability for peer-support, the group make-up including the sharing of past experiences, and in some contexts, how these groups are divided across age, gender and other social characteristics, as well as the facilitators’ characteristics, skill-level and own level of support.

The group format of GPFA increases the capacity of psychosocial support provision through providing PFA to several individuals at the same time, while also fostering support and relationship building between group members [43]. Additionally, because it is not required for facilitators to be mental health specialists, GPFA offers a practical approach to provide psychosocial support to populations affected by a crisis, particularly in low-resource settings, once the sufficient training of facilitators occurs [42,57]. That said, GPFA remains an intensive approach that needs comprehensive consideration prior to implementation.

GPFA is a complex approach that should be embedded within wider support systems. As such, linkages and well-structured supports are required for the successful implementation of GPFA. GPFA should therefore only be implemented when organisations can either link to or provide additional resources to participants, specifically basic needs support and further services (i.e., more advanced mental health support). Appropriate staff make-up and competencies are also essential, including the availability of supervisors, trainers and facilitators. Resources for facilitators should also be available. Therefore, organisations should consider building appropriate support systems to ensure the successful implementation and impact of GPFA. 

GPFA is applicable to a wide variety of contexts, including resource-constrained contexts. However, best practice still needs to be applied to ensure the ethical and appropriate support of participants. Our findings highlight many of the different contextual nuances necessary to consider in the implementation of GPFA. Of main importance among these findings is the recognition that GPFA is implemented in different contexts, and the specifics of GPFA are not one size fits all. The design or implementation of GPFA should therefore be preceded by a thorough contextual analysis which aims to identify: (1) the existing support services available for linkages and referral; (2) basic needs requirements and the ability of the organisation to support or provide these; (3) group history and experiences (e.g., are they a pre-existing group or to be newly formed); (4) socio-cultural conditions for the composition of the group, including any gender, age, or cultural considerations; and (5) the characteristics of facilitator(s) and how they will be trained, supervised and supported. Testing the finalised programme theories through real-world implementation would strengthen our understanding of how GPFA works, for whom, in different contexts. Additionally, the ways in which informal communication networks can be integrated and support GPFA members should be examined. Finally, exploring opportunities and challenges to provide GPFA remotely would be beneficial, especially in the context of epidemics such as Coronavirus Disease (COVID-19). 

### 4.1. Youth and GPFA

This review did not yield any literature regarding provision of GPFA for youth. Instead, the research team had to infer how GPFA could work with youth through examining literature about other group-based interventions commonly implemented with this age group. Evidence suggests that youth would be receptive and benefit from the group format of GPFA, given that group interventions and discussions are common practices within youth services [58,59,60,61], and that peer-based learning and group techniques are strongly aligned to support the developmental stages of this age group [59,60,62]. Receiving psychosocial care alongside peers can further reduce feelings of stigma associated with individual counselling [63]. Finally, the relationship building and sense of belonging that is fostered in group settings facilitates recovery among youth through developing interpersonal skills, working toward shared goals, and decreasing isolation [62,63].

### 4.2. Limitations 

The current study is not without limitations. First, there is an overall paucity of research on GPFA for humanitarian workers. Consequently, a large proportion of our preliminary documents consisted of implementation guidance, as opposed to research studies. As such, we included articles that more broadly applied to GPFA, such as those related to PFA, GPFA outside of humanitarian workers, and those that described other group-based events for humanitarian workers and first responders. Second, how GPFA was implemented varied across the included resources. Importantly, while the reference panel and expert group included individuals with experience in designing and implementing PFA/GPFA, humanitarian workers and volunteers who have participated in GPFA, and specifically youth, were not included. 

## 5. Conclusions

The humanitarian workforce faces many challenges, with staff and volunteers at an increased risk of anxiety, depression, and post-traumatic stress disorder. GPFA is widely recommended and implemented to provide humane, supportive and practical help in a group setting after an acute or during an ongoing event. However, there is a dearth of evidence on how, why and for whom GPFA works to address the needs of this cadre. What is more, the literature on GPFA for youth is extremely sparse. The current review puts forward a number of programme theories to advance our understanding of ‘how, why, for whom and in what contexts’ GPFA works. By applying these theories to existing evidence on youth, we have provided further key contextual and programmatic insights into GPFA for this specific demographic. Largely centring on the benefits of having appropriately implemented peer support, GPFA enables individuals to understand their natural reactions to stressful events and develop adaptive coping strategies, while also building social connections that promote a sense of belonging and security. The integrated design of GPFA ensures that individuals are linked to additional supports and have their basic needs addressed. While this approach is based on sparse evidence, its applicability to youth and its ability to provide support to humanitarian workers remains promising.

## Figures and Tables

**Figure 1 ijerph-18-01452-f001:**
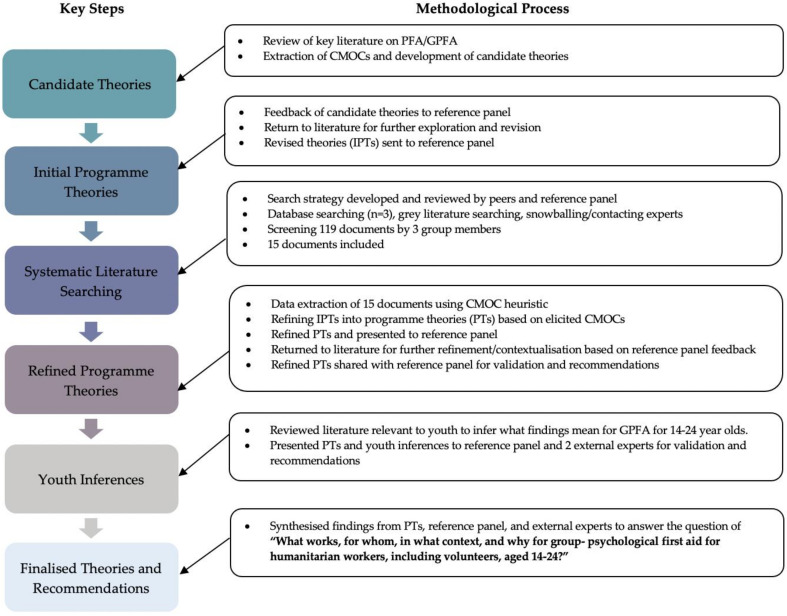
This figure highlights the key steps throughout the research process and an overview of the methods used. PFA: Psychological First Aid; GPFA: Group Psychological First Aid; CMOC: Content-Mechanism-Outcome Configuration; IPTs: Initial Programme Theories.

**Figure 2 ijerph-18-01452-f002:**
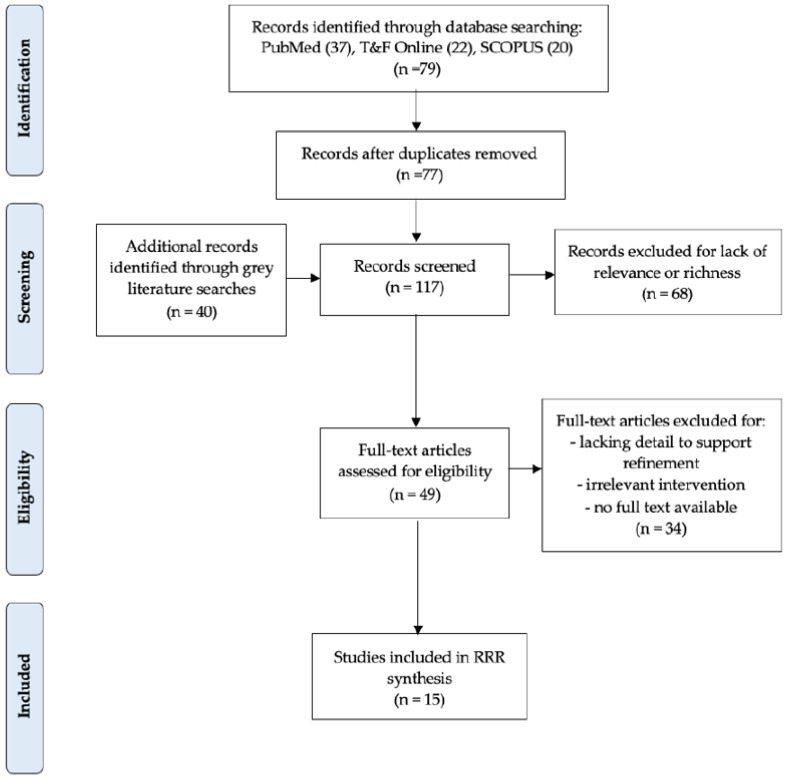
Modified PRISMA diagram.

**Figure 3 ijerph-18-01452-f003:**
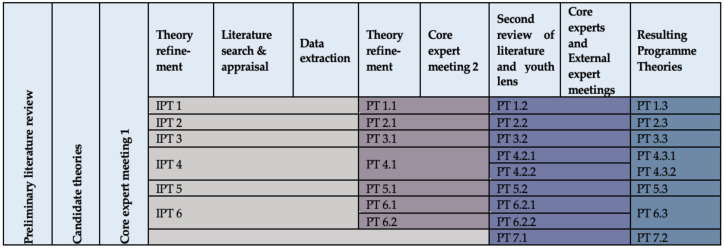
Timeline of the programme theory development.

## Data Availability

The data presented in this study are available in Appendix A: CMOCs and PT refinement.

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
