# Peer review of "A Rapid Realist Review of Group Psychological First Aid for Humanitarian Workers and Volunteers"

_ijerph, 2021, doi:10.3390/ijerph18041452_

Round 1
Reviewer 1 Report
This manuscript presents a rapid realist review conducted by synthesising grey literature and peer reviewed publications to develop programme theories related to the use of Group based Psychological First Aid (GPFA) interventions to reduce the impact and severity of exposure to traumatising experiences in humanitarian workers. An expert reference panel contributed to developing and refining the programme theories. Additionally, the use of GPFA for younger cohorts has been theorised drawing on literatures relating to older adults. This paper is very well written and presents an important and original contribution through the appropriate use of realist research to developing understanding of the key mechanisms that may underlie the successful application of an intervention for a population within which GPFA appears to be increasingly used.
I thank the authors for their obvious diligent work in conducting the review and preparing this manuscript. I have no significant concerns about this paper but believe a few minor amendments would further improve the already high standard of this article. Chiefly, the manuscript needs to be more explicit about the research process and written to be clearer to follow for readers who don’t have a grounding in realist research, as accounts of the steps taken are not always articulated (specific points where further detail would be beneficial are listed below).
Introduction
The paragraph starting on Line 45 gives only one reference for the initial point of a different impact on national and non-national aid workers. The Lopes Cardozo et al. paper was based solely on humanitarian aid workers in Kosovo, which was an extreme sectarian conflict, further the authors themselves acknowledged elements of the time at which that study was conducted (a year after NATO intervention) meant the findings may not be generalisable to workers in other contexts. As nationality is not a factor further expanded upon within the present manuscript I would suggest revising down this paragraph.
Line 86 (Please correct spelling of Hobfoll from ‘Hobfall’). A sentence or two explaining Hobfoll and colleagues’ work or how this led to the development of PFA, given it was first described by Schultz and Forbes, would be appreciated. Also, a further account for the widespread and increasing use of GPFA within this sector would further enhance the context described. A clear description of PFA is only presented in the discussion. I suggest moving Lines 387 – 396 to this point in the introduction. However, in relation to the authors comment, currently in the Discussion section Lines 387-391 about CISD, I gather this is a not uncontested point which forms part of a much broader debate that is not for this paper. As it is my understanding that PFA has closer links in origin to Critical Incident Stress Management CSIM, of which CSID can be a component, I would advise the authors to consider removing or broadening this sentence.
Line 101-102. I question the robustness of the assertion that exposure to trauma can be isolating within the context of humanitarian aid work in the way it is presented here. The reference given (Ulman) is out of numerical sequence and the clause doesn’t appear to contribute to the flow of the introduction, especially as the referenced manual doesn’t relate to groups of people but to individuals receiving treatment in a group setting.
Line 124 as presented it is unclear whether the authors are referring to young workers as one group and volunteers as a separate group not being considered in an age-related sense. Please consider moving the clause in parenthesis to the end of the sentence if this was not the intended meaning.
Materials and Methods
The authors have used the resource & response conceptualisation of mechanism in articulating the CMOCs presented in the appendix and once within the text. I do not consider it an issue that this is not consistently written, however, I would ask that some reference within the opening of this section to either Pawson’s original description of a Mechanism as Resource and Response, or Dalkin’s paper which expanded upon this concept.
Line 169 Please identify or at least enumerate the “several pieces of key literature” with which you started.
Line 185 should this be references not “referenced”?
Line 193 A little more detail on how the authors defined whether a source material contributed significantly to the PT development would be appreciated.
Results
Fig 2. Please indicate the reasons for exclusion of the 68 records removed at screening within the modified PRISMA diagram.
Table 1 the numbering of the developing PTs is a little confusing, I wonder whether indicating the iterative revision needs alteration to the numbers or if there is a different way to show this. It may be sufficient that it is specified that revision took place at each step.
Line 346 should this state power dynamics are not “is”
Finally, I wish to further commend the authors for their clear presentation of CMOCs and their process of PT development, as articulated in the main manuscript section 3.4 and the supplementary file. I look forward to citing this work in future.
Reviewer 2 Report
I appreciate the authors’ work on reviewing this important issue.
Strengths:
- A comprehensive summary of the information about group psychological first aids for humanitarian workers and volunteers.
- Propose a practical and feasible way to provide group psychological first aids for humanitarian workers and volunteers.
Limitations
- Lack of information for how to train therapists and the qualifications of therapists (the authors have included the lack of information in this aspect as one of the limitations).
The title could be shortened to “Group Psychological First Aid for Humanitarian Workers and Volunteers.”
Reviewer 3 Report
A systematic review of psychological first aid and useful in the treatment of humanitarian workers.
However, since the descriptions in Chapters 3.4.1 to 3.4.7 are difficult to read, please correct them in an easy-to-understand manner such as putting them together in a table.
Reviewer 4 Report
This paper is relevant in current times. Not only about the pandemical situation; of course, the humanitarian work it's continuing around the world working to attend to people who are living in war, climate, starving, and another kind of crisis in difficult contexts to develop their lives and personal capabilities.
Next, I mention some aspects to strengthen the manuscript.
1.1. The Problem
-
Briefly explain the intervention contexts where they carry out their labor, including citations.
- Line 50-55: I would recommend an argument or contribution of a theory that explains this. including citations.
- Line 57-58: Support this point with more research, since it is the axis that justifies this work. In case there are not, reflect it.
- Line 59-61: "younger humanitarian aid workers and volunteers": explain if they are expatriates, national staff, or both.
- Line 61-64: It is advisable to explain this argument with the support of other previous research.
- Line 66-68: More research to support this claim.
1.2. Group Psychological First Aid
Briefly introduce information on how the degree of experience of a traumatic experience is evaluated in the context of humanitarian aid. That is, in whose hands is it left for the humanitarian worker/volunteer to participate in the PFA or GPFA?
- Line 93-94: To cite the source of this statement.
- Line 113-119: What were the parameters to understand them as “comparable”? Explaining through a figure would be very clarifying.
2.1. Reference Group and Experts
- Line 160-163: In what kind of contexts and humanitarian work: natural disasters, war, disease…?
- Line 163-165: Which are the most recognized?
2.2. Preliminary Review of the Literature and Initial Programme Theories
- Line 169: Include citations.
2.3. Changes to Process - Youth Inferences
Which inferences? Explain maybe with another figure or table.
3.2. Defining Group Psychological First Aid
- Line 226-227: What differences? Include an explanatory figure providing PFA and GPFA data.
- Line 230: What does the scientific literature say? Include and cite any definition that supports this one that is presented.
- Line 240-242: Several times a week, fortnightly? What does the scientific literature say?. Remember to address How are these cases detected, how are these values measured, who would do it…. Zoom in briefly on the introduction.
- Line 245: sex or gender? Gender includes cultural properties.
3.4.2. Programme Theory 2: Meeting Basic Needs
Include the definitions of those concepts: basic physical and psychological needs of participants, including citations.
- Line 280-281: Include citations.
3.4.6. Programme Theory 6: Sustainability and Accessibility
- Line 357-359: Include citations.
